# Evidence for a common evolutionary rate in metazoan transcriptional networks

**Anne-Ruxandra Carvunis†, Tina Wang†, Dylan Skola†, Alice Yu, Jonathan Chen, Jason F Kreisberg, Trey Ideker\***

Department of Medicine, University of California, San Diego, La Jolla, United States

**Abstract** Genome sequences diverge more rapidly in mammals than in other animal lineages, such as birds or insects. However, the effect of this rapid divergence on transcriptional evolution remains unclear. Recent reports have indicated a faster divergence of transcription factor binding in mammals than in insects, but others found the reverse for mRNA expression. Here, we show that these conflicting interpretations resulted from differing methodologies. We performed an integrated analysis of transcriptional network evolution by examining mRNA expression, transcription factor binding and *cis*-regulatory motifs across >25 animal species, including mammals, birds and insects. Strikingly, we found that transcriptional networks evolve at a common rate across the three animal lineages. Furthermore, differences in rates of genome divergence were greatly reduced when restricting comparisons to chromatin-accessible sequences. The evolution of transcription is thus decoupled from the global rate of genome sequence evolution, suggesting that a small fraction of the genome regulates transcription.

**\*For correspondence:** tideker@ucsd.edu

†These authors contributed equally to this work

**Competing interests:** The authors declare that no competing interests exist.

## Introduction

A long-standing question in biology is what fraction of the genome regulates transcription (*ENCODE Project Consortium, 2012*; *Graur et al., 2013*; *Niu and Jiang, 2013*; *Kellis et al., 2014*). Recent studies of chromatin structure have implicated half of the human genome in regulatory interactions (*ENCODE Project Consortium, 2012*). Comparative genomic studies, however, have shown that less than 10% of the human genome is evolutionarily conserved (*Siepel et al., 2005*), suggesting that many of the experimentally-detected interactions are not functional (*Graur et al., 2013*). Recent studies have measured the association between sequence changes and changes in transcript levels, epigenetic modifications or binding of transcription factors regulating specific gene sets (gene-specific transcription factors, GSTF) (*Cookson et al., 2009*; *McVicker et al., 2013*; *Kasowski et al., 2010*; *2013*; *Heinz et al., 2013*; *Villar et al., 2014*; *Wong et al., 2015*; *Brem et al., 2002*; *Chan et al., 2009*; *Shibata et al., 2012*). These experiments demonstrated that genomic sequences can influence transcription even in the absence of evolutionary conservation. For instance, some repetitive elements previously thought to be 'junk' DNA have been shown to effectively regulate gene expression (*Rebollo et al., 2012*). The rapid evolution of repetitive and other rapidly-evolving sequences could cause pervasive rewiring of transcriptional networks through creation and destruction of regulatory motifs (*Villar et al., 2014*). Such rapid transcriptional evolution would set mammals apart from other metazoans like birds or insects, whose genomes contain far fewer repetitive elements (*Taft et al., 2007*) and tend to be more constrained (*Siepel et al., 2005*; *Zhang et al., 2014*).

A few studies have attempted to assess whether transcriptional networks evolve more rapidly in mammals than in insects from the fruit fly genus *Drosophila*. These studies have reached conflicting conclusions. When examining the evolution of GSTF binding, chromatin immuno-precipitation (ChIP) studies in mammalian livers have generally described faster divergence rates than similar studies in

**eLife digest** The genetic information that makes each individual unique is encoded in DNA molecules. Cells read this molecular instruction manual by a process called transcription, in which proteins called transcription factors bind to DNA in specific places and regulate which sections of the DNA will be expressed. These 'transcripts' are active molecules that determine the cell's – and ultimately the individual's – characteristics. However, it is not well understood how alterations in the DNA of different individuals or species can lead to changes in where the transcription factors bind, and in which transcripts are expressed.

Carvunis, Wang, Skola et al. set out to determine if there is a relationship between how often DNA changes and how often transcription changes during the evolution of animals. The experiments examined the abundance of transcripts in the cells of a variety of animal species with close or distant evolutionary relationships. For example, the house mouse was compared to a close relative called the Algerian mouse, to another species of rodent (rat) and to humans.

The experiments show that the changes in transcript abundances are happening at similar rates in mammals, birds and insects, even though DNA changes at very different rates in these groups of animals. This similarity was also observed for other aspects of transcription, such as in changes to where transcription factors bind to DNA. The next challenges are to find out what makes transcription evolve at such similar rates in these groups of animals, and whether these findings extend to other species and to other processes in cells.

fly embryos (*Villar et al., 2014*; *Stefflova et al., 2013*). However, divergence rates were estimated with different analytical methods in the different ChIP studies (*Supplementary file 1*) (*Villar et al., 2014*; *Bardet et al., 2012*). Another study found that gene expression levels may diverge at a slower rate in mammals than in flies, by comparing genome-wide correlations of mRNA abundances estimated by RNA sequencing (RNA-seq) for mammals but by a mixture of technologies for flies including microarrays (*Coolon et al., 2014*). Although the inconsistencies between these conclusions may indicate that the evolution of transcriptional networks is fundamentally different in mammals and insects, they may also reflect a sensitivity of evolutionary rate estimations to technical methodology.

Here, we jointly examined the evolution of gene expression levels and the underlying genome-wide changes in GSTF binding and *cis*-regulatory sequences using consistent methodologies both within and across various animal lineages.

## Results

We assembled a comparative genomics platform encompassing >40 publicly available datasets spanning >25 organisms representative of the *Mammalia* (mammals), *Aves* (birds) and *Insecta* (insects) phylogenetic classes (*Figure 1—figure supplement 1*). We designed a statistical framework to objectively compare the rates of divergence of these various datasets across lineages. In brief, an exponential model describing evolutionary divergence under a common, lineage-naïve rate was evaluated against a lineage-aware model, accounting for both statistical significance and effect size (*Figure 1*; Materials and methods). We assessed the power of this statistical framework using simulations and found that it could detect differences in divergence rates with high sensitivity (Materials and methods; *Figure 1—figure supplement 2*).

As a baseline, we first performed a comparative analysis of the evolution of genome sequences. We randomly sampled genomic segments from designated reference genomes: *Mus musculus domesticus* (C57BL/6) for mammals, *Gallus gallus* for birds and *Drosophila melanogaster* for insects. The rates at which genomic segments that retained homologs with the other species within each lineage accumulate nucleotide substitutions were then estimated and compared using our statistical framework. Segments retaining homologs displayed high sequence conservation across all three lineages, although our framework detected a slightly but significantly faster divergence in insects than in mammals or birds ($P<0.05$; *Figure 2—figure supplement 1*). Next, we compared the rates at which randomly sampled genomic segments lost homology with the other species within each lineage. We observed a much larger difference in evolutionary rates across lineages using this measure

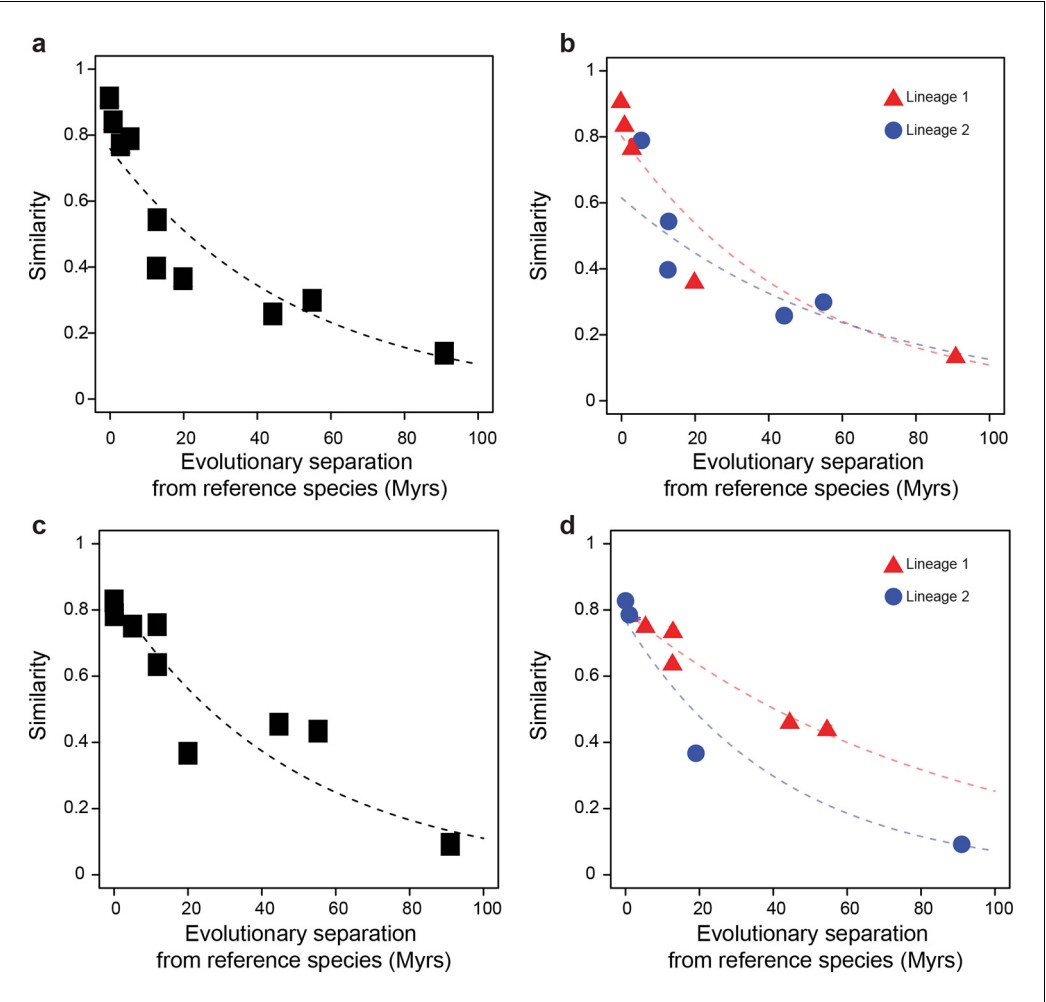

**Figure 1.** Statistical framework to evaluate differences in evolutionary rates of change. Throughout this study we frequently evaluated whether the rate of evolutionary divergence of a given layer of transcriptional regulation differs between lineages. Our approach is equivalent to asking: if the lineage labels were hidden, would one be able to tell that the data points correspond to several lineages or would they seem equally likely to belong to a common distribution? (a, b) Depict an example of statistically indistinguishable evolutionary rates. Without lineage labels (a), the similarity data are modeled by an exponential decay as well as with lineage labels (b). Adding lineage labels does not significantly improve the fit. (c, d) Depict an example of statistically different evolutionary rates. Adding lineage labels (d) significantly improves the fit of an exponential decay model over unlabeled data (c).

The following figure supplements are available for figure 1:

**Figure supplement 1.** Comparative genomics platform for studying transcriptional network evolution across three metazoan lineages.

**Figure supplement 2.** Power of the statistical framework to evaluate differences in evolutionary rates.

---

($P<0.05$; *Figure 2*; *Figure 2—figure supplement 2*). For instance, after 100 million years (Myrs) of evolution, only ~30% of mammalian segments retained homology, whereas >60% of bird and insect segments did. These findings recapitulated previous observations according to which genome sequences are less constrained in mammals than in insects (*Siepel et al., 2005*) or birds (*Zhang et al., 2014*).

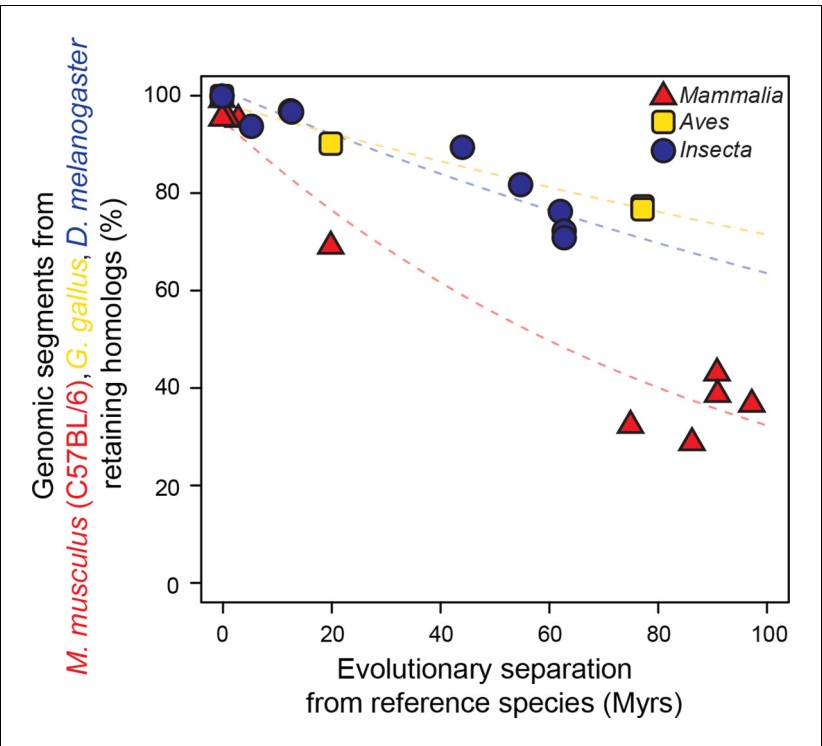

**Figure 2.** Genomic sequences evolve more rapidly in mammals than in birds and insects. The evolutionary retention of 5000 randomly sampled 75 bp segments was averaged over 20 trials. Organisms compared to reference species are as follows: *M. musculus domesticus* (AJ), *M. musculus castaneus, M. spretus*, rat, guinea pig, rabbit, human, chimpanzee and dog for *Mammalia*; turkey, zebrafinch and flycatcher for *Aves*; *D. simulans, D. erecta, D. yakuba, D. ananassae, D. pseudoobscura, D. virilis, D. willistoni* and *D. grimshawi* for *Insecta*. Colored dashed lines: lineage-specific exponential fits, here and in all following displays. The trends were robust to variations in segment length and sequence similarity filters (*Figure 2—figure supplement 2*).

The following figure supplements are available for figure 2:

**Figure supplement 1.** Genomic segments retaining homologs are highly conserved at the nucleotide level.

**Figure supplement 2.** Retention of genomic segments is robust to changes in sampled region size and sequence identity threshold.

We then studied the evolution of gene expression levels, using exclusively RNA-seq datasets. In mammals and birds, these datasets were generated from adult livers; in insects, they were from whole bodies of adult female fruit flies (Materials and methods; *Figure 3—source data 1*). After determining expression levels for each gene in each species using a common data processing pipeline, we correlated the expression levels of genes in the reference species with the expression levels of their one-to-one orthologs in all other species within the same lineage (Materials and methods). We found that correlations of gene expression levels decreased over time at similar rates that were statistically indistinguishable: a lineage-naïve model describing the evolution of gene expression levels under a common rate fitted the data as well as a lineage-aware model (*Figure 3*). This result was robust to changes in correlation metrics or inclusion/exclusion of poorly expressed genes (*Figure 3— figure supplement 1*).

Several lines of evidence suggest that gene expression levels can remain relatively stable even as the genomic locations bound by GSTFs change rapidly over time (*Wong et al., 2015*; *Chan et al., 2009*; *Paris et al., 2013*). Therefore, we next examined the evolution of GSTF binding patterns. We considered all GSTFs that were profiled using ChIP followed by massively parallel sequencing (ChIP-seq) in at least three related species, where separate ChIPs were performed per species. GSTFs

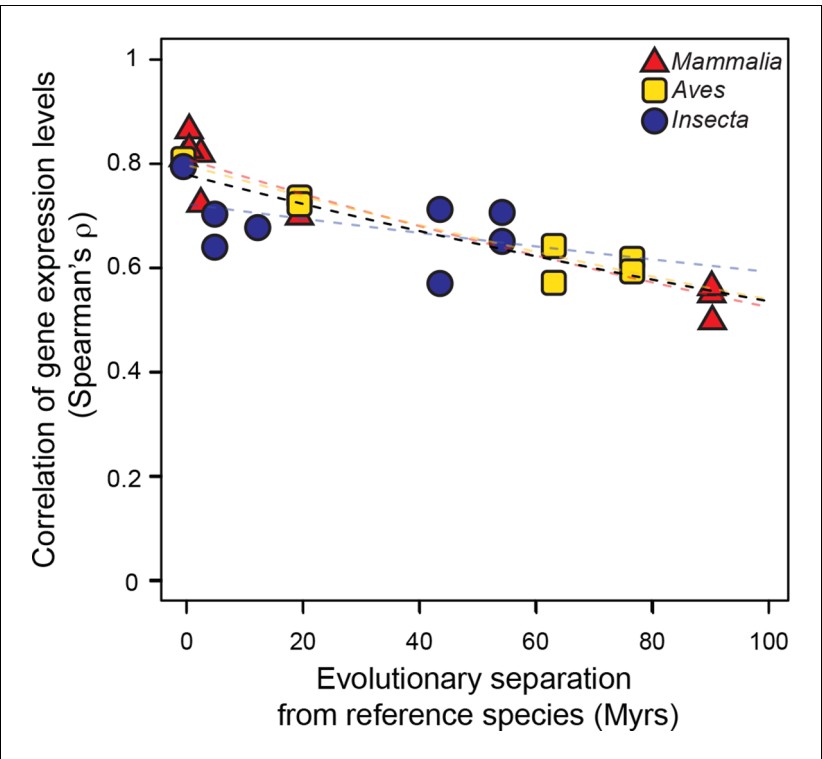

**Figure 3.** Gene expression levels diverge at a common rate in mammals, birds and insects. Gene expression levels were derived independently from two RNA-seq experiments for each reference species and then correlated against each other and against gene expression levels derived from individual experiments in other species within the same lineage. Black dashed line: lineage-naïve exponential fit of all the data, without differentiating the lineages, here and in all following displays. Organisms compared to reference species are as follows: *M. musculus castaneus, M. spretus*, rat, human and gorilla for *Mammalia*; turkey, duck and flycatcher for *Aves*; *D. simulans, D. yakuba, D. ananassae* and *D. pseudoobscura* for *Insecta*.

The following source data and figure supplement are available for figure 3:

**Source data 1.** Accession numbers used in RNA-seq analyses.

**Figure supplement 1.** The common evolutionary rate of gene expression levels presented in *Figure 3* is robust to changes in correlation metrics or expression threshold.

meeting these requirements were Twist and Giant in fruit fly embryos, and CEBPA, FOXA1 and HNF4A in mammalian livers (Materials and methods; *Figure 4—source data 1*; *Supplementary file 1*). We aimed to measure cross-species similarity in GSTF occupancy with a unified analytical method across all of these datasets. Despite the widespread use of ChIP-seq, there is no consensus on the appropriate analytical method (*Wilbanks and Facciotti, 2010*). ChIP-seq analysis pipelines typically discretize continuous occupancy profiles into a set of occupied segments ('peaks'), but this step requires choosing a signal processing algorithm (a peak caller) and associated parameters (*Figure 4a*). Further comparison of occupied segments across species requires additional analytical choices (*Figure 4a*), some of which can strongly influence downstream findings (*Bardet et al., 2012*).

To explore the impact of these choices, we processed all ChIP-seq data using systematic combinations of parameters representative of, and expanding from, previous studies (*Supplementary file 1*) (*Landt et al., 2012*). In total, we executed 108 analytical pipelines to compare divergence rates across 6 pairs of GSTFs (2 in insects each compared with 3 in mammals), the occupancy profiles of which were examined in 3–7 species per lineage (Materials and methods). The values of the estimated rates varied greatly from one combination of parameters to the next (*Figure 4b,c*). However, in the majority of cases (56–78% over the 6 comparisons), GSTF binding patterns diverged at

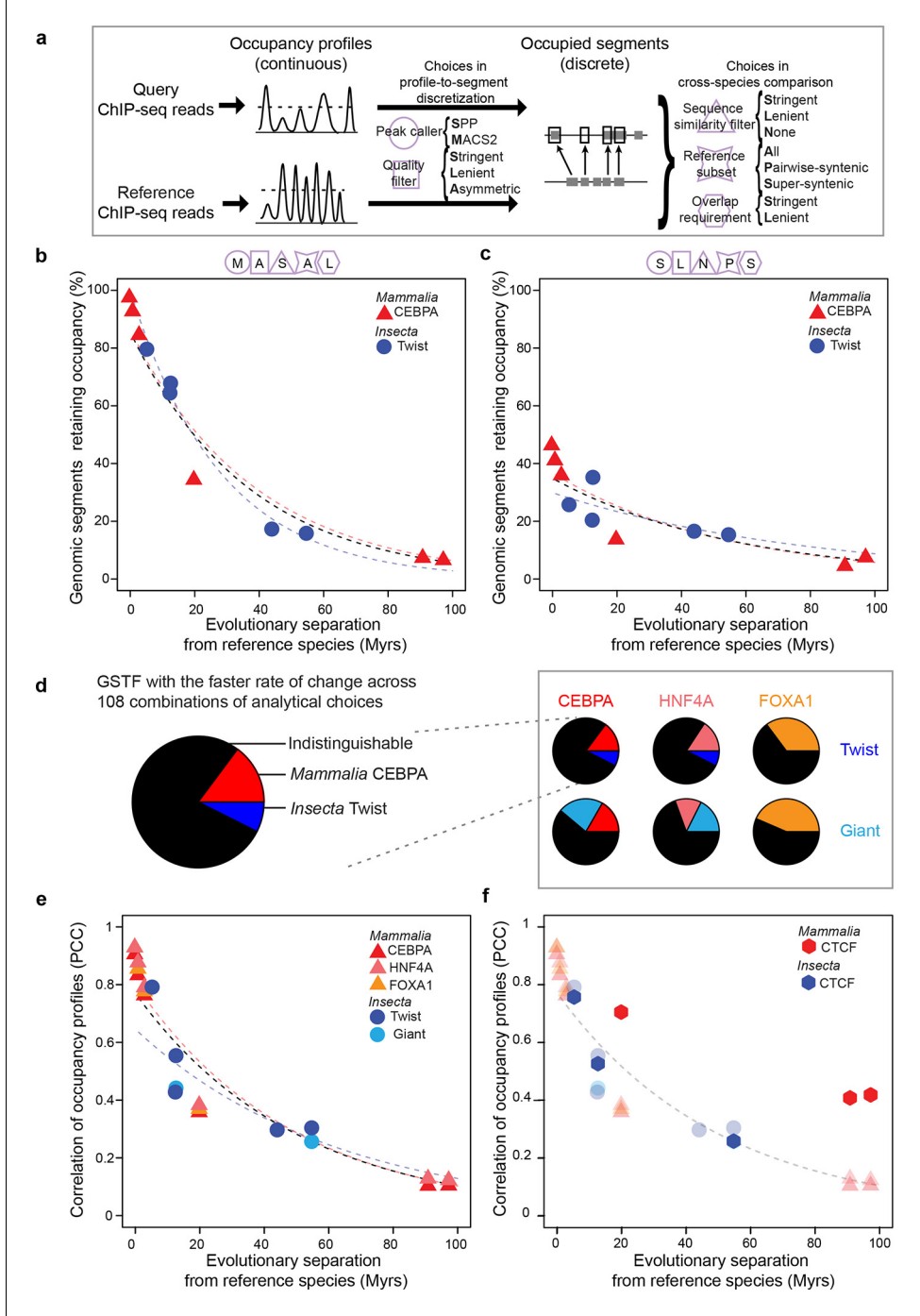

**Figure 4.** GSTF occupancy diverges at a common rate in mammals and insects. (**a**) Estimating shared GSTF occupancy across species requires multiple parameter choices. This diagram summarizes the main steps involved in comparing GSTF-occupied segments across species, showing a representative sample of choices at each step (steps represented by purple shapes, specific choices by the first letter bolded). The detailed methods and specific choices illustrated here and implemented in panels b–d are described in Materials and methods. (**b, c**) An example of different analytical choices leading to different results despite starting from the same underlying data. Organisms compared to reference species are as follows: *M. musculus domesticus* (AJ), *M. musculus castaneus*, *M. spretus*, rat, human and dog for *Mammalia; D. simulans, D. erecta, D. yakuba, D. ananassae* and *D. pseudoobscura* for *Insecta*. (**d**) Most combinations of choices yield indistinguishable evolutionary rates of GSTF binding patterns across lineages. The comparison of Twist and CEBPA is enlarged to show the color labels corresponding to the statistical interpretation regarding relative evolutionary rates. (**e**) A genome-wide comparison

*Figure 4 continued on next page*

*Figure 4 continued*
of GSTF occupancy profiles at single-nucleotide resolution shows indistinguishable evolutionary rates for CEBPA, HNF4A and FOXA1 in mammals, and for Twist and Giant in insects. PCC: Pearson correlation coefficient. (f) CTCF occupancy is highly conserved in mammals. Transparent points and lines are identical as in panel **e**. Hexagons correspond to cross-species correlations of CTCF occupancy at single-nucleotide resolution.
The following source data and figure supplement are available for figure 4:

**Source data 1.** Accession numbers used in ChIP-seq analyses.
**Source data 2.** 648 segment-based ChIP analyses.
**Figure supplement 1.** Measured GSTF binding divergence rates are influenced by parameter choices.

statistically indistinguishable rates in mammals and insects (*Figure 4d*; *Figure 4—source data 2*). Although the computed divergence rates were sensitive to technical methodology (*Figure 4—figure supplement 1*), for a given method the results were generally similar across lineages for all of the five GSTFs investigated.

To substantiate these findings, we devised a method to compare genome-wide occupancy profiles at single-nucleotide resolution without discretization. We correlated occupancy profiles between pairs of species across all nucleotides where genomes aligned, after accounting for the differences in sequencing depth, read length and fragment size across datasets (Materials and methods). Again, we found indistinguishable divergence rates, regardless of which GSTF or lineage was examined (*Figure 4e*). After 100 Myrs of evolution, the correlation of GSTF occupancy profiles was 0.10 in mammals and 0.13 in insects. As a control, we also applied this method to CTCF, a pleiotropic DNA-binding protein that acts as chromatin insulator and looping factor (*Ohlsson et al., 2010*). In mammals, patterns of DNA occupancy have been shown to be more conserved for CTCF than for GSTFs using unified analytical methods (*Schmidt et al., 2012*). In contrast, CTCF DNA occupancy was shown to diverge rapidly in insects, perhaps due to the existence of other insulator proteins (*Villar et al., 2014*; *Ni et al., 2012*). Our analysis successfully recapitulated this difference (*Figure 4f*), demonstrating that the common evolutionary rate observed among GSTFs (*Figure 4e*) was not an artifact of our method for profile correlation.

The similarity of divergence rates observed across lineages for gene expression levels (*Figure 3*) and GSTF binding patterns (*Figure 4*) was unexpected given the rapid evolution of genomic sequences in mammals relative to insects (*Siepel et al., 2005*) or birds (*Zhang et al., 2014*) (*Figure 2*). We therefore further examined these trends at the level of *cis*-regulatory sequences. First, we considered the DNA sequence motifs thought to be specifically recognized by the mammalian and insect GSTFs included in the previous ChIP-seq analysis (*Figure 4*). We identified locations with significant matches to these motifs throughout the genomes of the reference species and estimated how frequently these loci retained the same motifs relative to background expectations (Materials and methods). We found similar, indistinguishable retention rates in mammals and insects (*Figure 5a*). Next, we studied the evolution of a broader set of motifs corresponding to GSTFs shared between *M. musculus* and *D. melanogaster*. We found that these motifs were retained at similar rates across lineages relative to background expectations in 8 out of 12 cases (one example shown in *Figure 5b*; all other cases in *Figure 5—figure supplement 1*).

Most active *cis*-regulatory sequences are located in genomic regions with accessible chromatin (*Hesselberth et al., 2009*). A recent study showed that chromatin-accessible sequences were significantly more conserved between human and mouse than expected by chance (*Yue et al., 2014*). We expanded this analysis to a wide range of species by using chromatin-accessible sequences identified by DNAse I hypersensitivity in *M. musculus* livers, *D. melanogaster* embryos and *G. gallus* MSB-1 cells (Materials and methods). We performed the segment sampling procedure described previously (*Figure 2*), after excluding genes and promoter regions since they typically are highly conserved (Materials and methods). Whereas inaccessible segments lost homology much faster in mammals than in insects and birds ($P<0.05$; *Figure 5c*), accessible segments retained homologs at more similar rates in the three lineages (*Figure 5d*; *Figure 5—figure supplement 2*). We still

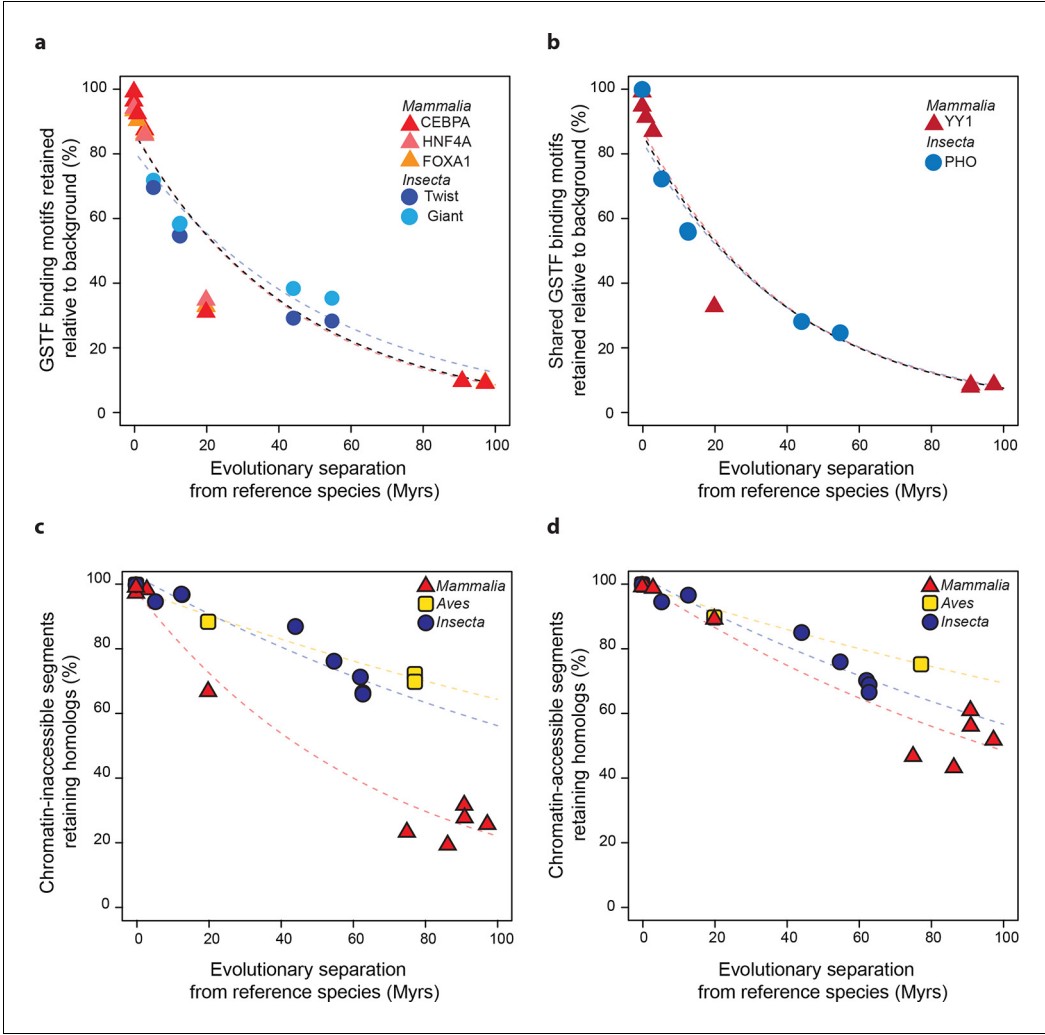

**Figure 5.** Regulatory sequences diverge at similar rates across lineages. (**a**) The motifs for CEBPA, HNF4A and FOXA1 in mammals and for Twist and Giant in insects are retained at a common rate. Organisms compared to reference species are the same as *Figure 4*. (**b**) The motifs for GSTFs shared in mammals and insects are retained at common rates. One example is shown here for the motifs corresponding to PHO (FBgn0002521) in *D. melanogaster* and YY1 (ENSMUSG00000021264) in *M. musculus*, which are orthologous GSTFs. Eleven other cases of motif evolution for shared GSTFs conserved in mammals and insects are shown in *Figure 5—figure supplement 1*. Organisms compared to reference species are as in *Figure 4*. (**c, d**) Chromatin-accessible sequences are retained at similar rates in mammals, birds and insects. Analyses were performed as in *Figure 2*, limiting sampling to the inaccessible (**c**) and accessible (**d**) portions of the intergenic regions. Organisms compared to reference species are the same as *Figure 2*. The trends were robust to variations in segment length and sequence similarity filters (*Figure 5—figure supplement 2*).

The following figure supplements are available for figure 5:

**Figure supplement 1.** Conservation of *cis*-regulatory motifs for GSTFs conserved across insects and mammals.

**Figure supplement 2.** Retention of intergenic genomic segments in accessible and inaccessible chromatin is robust to changes in sampled region size and sequence identity threshold.

detected statistically significant differences across lineages (*P*<0.05), but the effect sizes were considerably smaller than for inaccessible segments. For instance, ~60% of segments retained homology after 100 Myrs in birds and insects, independently of accessibility, whereas ~50% of chromatin-accessible segments and only ~20% of inaccessible segments did so in mammals.

## Discussion

To our knowledge, the analyses presented here represent the most comprehensive study conducted to date on the evolution of transcriptional networks across animal lineages. By applying unified analytical methods to data from different lineages, we were able to glean novel insights into the evolution of transcription in animals. We observed that gene expression levels, GSTF binding patterns, regulatory motifs and chromatin-accessible sequences each diverged at rates that were similar across mammals, birds and insects. These unexpected results reconcile previously conflicting findings (*Villar et al., 2014*; *Coolon et al., 2014*), highlighting the importance of unified study methodologies and providing evidence for a common evolutionary rate in metazoan transcriptional networks.

Most functional genomics studies have focused on humans and model organisms such as *D. melanogaster* or *M. musculus*, which are distantly related to each other. However, data on closely related species, like those which we collected in this study, are needed to investigate the dynamics of molecular network evolution. Unfortunately, such data remain scarce, leading to important limitations of our work. We only investigated three lineages and six to twelve organisms per lineage with non-uniform coverage over evolutionary time. In addition, we only examined a small number of tissues for each lineage and a total of five GSTFs (none in birds). The generalizability of our observations thus remains to be further evaluated as more data becomes available. Despite these limitations, our finding that transcriptional networks evolve at a common rate per year across animal lineages was strikingly robust across data layers.

The underlying mechanisms responsible for this concordance of evolutionary rates are unclear. Mammals, birds and insects exhibit wide differences in the features that are traditionally associated with evolutionary rates, such as generation times and breeding sizes. Populations with small breeding sizes, such as mammals, are thought to be more prone to genetic drift (*Ohta, 1992*). This theory accounts for the abundance of repetitive elements and the rapid evolution of genomic sequences in mammals relative to insects, which have much larger breeding sizes. If the same theoretical principles also governed the evolution of transcriptional networks, we would have expected that transcription would evolve more rapidly in mammals than in insects. Instead, our results show that the evolution of transcriptional networks, whether slow (e.g. transcript levels) or fast (e.g. GSTF binding), is decoupled from the lineage-specific features that govern genome sequence evolution.

One potential model could be that repetitive and rapidly-evolving sequences, which make up the majority of the mammalian genome (*Siepel et al., 2005*; *Taft et al., 2007*), play a negligible role in the global regulation of gene expression. Rather, chromatin-accessible regions may represent the only portion of the mammalian genome that effectively regulates transcription. We observed that chromatin-accessible regions diverge much more slowly than other non-coding sequences in mammals, consistent with previous findings (*Yue et al., 2014*). These differences in divergence rates, however, were not found in birds and insects. As a result, chromatin-accessible regions in mammals are conserved at levels similar to those in birds and insects, in contrast to the genome as a whole. According to this model, the similar rates of evolution of chromatin-accessible sequences would constrain the dynamics of transcriptional evolution to be similar across lineages. The regulatory potential of repetitive and other rapidly-evolving elements could be rendered functionally inconsequential by silencing, or could be concentrated on controlling the expression of genetic elements that we did not investigate, such as non-coding RNAs or species-specific genes (*Sundaram et al., 2014*).

An alternative model could be that the sequences that control transcriptional regulation in birds and insects evolve particularly rapidly within otherwise stable genomes. In these organisms, transcriptional networks would diverge under the action of natural selection, through specific single nucleotide substitutions resulting in rapid compensatory turnover (*He et al., 2011a*). In mammals, transcriptional networks would diverge in a largely neutral fashion driven for instance by transposable elements (*Sundaram et al., 2014*). In this case, similar rates of transcriptional divergence across lineages would arise through very different evolutionary processes.

Importantly, none of the aforementioned models account for the differences in generation times between lineages. Evolutionary changes occurring based on chronological time and not generation time has also been observed for many protein-coding sequences. Observations such as these led to the molecular clock theory (*Kumar, 2005*). The mechanisms through which environmental forces entrain these chronological evolutionary clocks remain to be elucidated (*Kumar, 2005*).

## Materials and methods

### Genome and annotation sources

We downloaded genome sequences for organisms belonging to three metazoan lineages: mammals, birds and insects. The mammalian and insect genome sequences were downloaded from the UCSC Genome Bioinformatics website (*Rosenbloom et al., 2015*): mm9 for *Mus musculus domesticus*, rn5 for *Rattus norvegicus* and hg19 for *Homo sapiens*; dm3 for *Drosophila melanogaster*, droSim1 for *Drosophila simulans*, droEre2 for *Drosophila erecta*, droYak2 for *Drosophila yakuba*, droAna3 for *Drosophila ananassae* and dp4 for *Drosophila pseudoobscura*. Genomes for mice strains and species not available from the UCSC Genome Bioinformatics site (*M. musculus domesticus* [AJ], *M. musculus castaneus* and *M. spretus*) were downloaded from (*Stefflova et al., 2013*). We downloaded bird genome sequences from Ensembl version 80 BioMart (*Cunningham et al., 2015*): galGal4 for *Gallus gallus*, Turkey_2.01 for *Meleagris gallopavo*, taeGut3.2.4 for *Taeniopygia guttata* and FicAlb_1.4 for *Ficedula albicollis*. Protein-coding gene names and symbols along with associated transcripts sequences were obtained from FlyBase (*dos Santos et al., 2015*) for insect species (dmel-r5.46, dsim-r1.4, dere-r1.3, dyak-r1.3, dana-r1.3 and dpse-r2.30), from Ensembl version 80 BioMart for bird species and from Ensembl version 59 BioMart for mammalian species (*Cunningham et al., 2015*). For *M. spretus* and *M. musculus castaneus*, we used the same transcript annotations as for *M. musculus*. Within the genomes of our designated reference organisms (*M. musculus domesticus, G. gallus* and *D. melanogaster*), we defined promoters as the region 0-2 kb upstream of transcription start site and delineated intergenic regions as regions that did not overlap annotated genes or promoters. Chromatin accessibility tracks used in *Figure 5c,d* and *Figure 5—figure supplement 2* were downloaded from the UCSC bioinformatics website (*Rosenbloom et al., 2015*) for *M. musculus domesticus* and *D. melanogaster*, and obtained from (*He et al., 2014*) for *G. gallus*. We restricted our analyses to the sequences or annotations in, or homologous to, the well-defined chromosome scaffolds of the reference organism. Specific reference chromosomes analyzed are as follows: *G. gallus* (1–28, Z, W), *D. melanogaster* (2L, 2R, 3L, 3R, 4, X) and *M. musculus* (1–19, X, Y).

### Homology and evolutionary relationships

We obtained orthology relationships between protein-coding genes using Ensembl COMPARA (*Vilella et al., 2009*), matching the Ensembl versions used for protein coding genes for each species described above. These relationships were used in *Figure 3*, *Figure 3—figure supplement 1*, *Figure 5b* and *Figure 5—figure supplement 1*. Homology between genomic segments was assigned using the LiftOver tool (*Rosenbloom et al., 2015*), for all analyses presented in *Figures 2*, *4* and *5* and associated figure supplements, with the exception of the nucleotide-resolution analysis of GSTF occupancy profiles presented in *Figure 4e,f*. We used pre-computed chain files from UCSC matching the genome versions listed above when chains were readily available (*Rosenbloom et al., 2015*). When chain files were not available, we built chain files to map the UCSC M. *musculus* C57BL/6 mm9 to the genomes of *M. musculus domesticus* AJ, *Mus musculus castaneus* and *Mus spretus*, as well as to map the Ensembl 80 galGal4 to the genomes of *M. gallopavo, F. albicollis* and *T. guttata* (*Figure 1—figure supplement 1*). These chains were constructed by following the steps recommended by UCSC (*Supplementary file 2*) (http://genomewiki.ucsc.edu/index.php/Whole_genome_alignment_howto).

For the nucleotide-resolution analysis of GSTF occupancy profiles, we assigned homology relationships using the chain files, or, in the case of mice strains, using genome mapping tables from (*Stefflova et al., 2013*). We filtered the chain files to obtain one-to-one unambiguous mappings by retaining only highest scoring alignment for each position. These filtered mappings were then used to transfer data to from any organism onto the corresponding reference genome. Regions in the reference species genome lacking one-to-one unambiguous mappings were excluded from analysis.

To define evolutionary distances separating species in Myrs, we chose published estimates generated as homogenously as possible within each lineage using a combination of sequence alignments and fossil records. All distances between insect species were taken from (*Tamura et al., 2004*); all distances between bird species were taken from (*Lu et al., 2015*); distances between mammalian species were taken from (*Stefflova et al., 2013*) and TimeTree (*Hedges, 2009*).

## Data sources

For RNA-seq analyses (*Figure 3*; *Figure 3—figure supplement 1*), sequencing data for the reference species corresponding to two experiments performed independently by different research groups, and, when possible, representing different genotypes, were downloaded from public repositories. For *M. musculus domesticus*, we used data from (*Goncalves et al., 2012*; *Sugathan and Waxman, 2013*), for *G. gallus* we used data from (*Brawand et al., 2011*) and (*Coble et al., 2014*), for *D. melanogaster* we used data from (*ENCODE Project Consortium, 2012*; *Chen et al., 2014*). Other species included were *M. musculus castaneus* (*Goncalves et al., 2012*), *M. spretus* (*Wong et al., 2015*), *R. norvegicus* (*Gong et al., 2014*), *H. sapiens* (*ENCODE Project Consortium, 2012*; *Lin et al., 2014*), *G. gorilla* (*Brawand et al., 2011*), *D. simulans* (*Chen et al., 2014*), *D. yakuba* (*Chen et al., 2014*), *D. ananassae* (*Chen et al., 2014*), *D. pseudoobscura* (*Chen et al., 2014*), *M. gallopavo* (*Monson et al., 2014*), *A. platyrhynchos* (*Huang et al., 2013*) and *F. albicollis* (*Uebbing et al., 2013*). Specific accession numbers are listed in *Figure 3—source data 1*.

For ChIP-seq analyses (*Figure 4*), we downloaded data for FOXA1 in *M. musculus domesticus* (C57BL/6) (*Stefflova et al., 2013*), *M. musculus domesticus* (AJ) (*Stefflova et al., 2013*), *M. musculus castaneus* (*Stefflova et al., 2013*), *M. spretus* (*Stefflova et al., 2013*) and *R. norvegicus* (*Stefflova et al., 2013*); HNF4A and CEBPA in *M. musculus domesticus* (C57BL/6) (*Stefflova et al., 2013*), *M. musculus domesticus* (AJ) (*Stefflova et al., 2013*), *M. musculus castaneus* (*Stefflova et al., 2013*), *M. spretus* (*Stefflova et al., 2013*), *R. norvegicus* (*Stefflova et al., 2013*), *H. sapiens* (*Schmidt et al., 2010*) and *C. familiaris* (*Schmidt et al., 2010*); Twist in *D. melanogaster* (*He et al., 2011b*), *D. simulans* (*He et al., 2011b*), *D. erecta* (*He et al., 2011b*), *D. yakuba* (*He et al., 2011b*), *D. ananassae* (*He et al., 2011b*) and *D. pseudoobscura* (*He et al., 2011b*); Giant *in D. melanogaster* (*Paris et al., 2013*; *Bradley et al., 2010*), *D. yakuba* (*Bradley et al., 2010*) and *D. pseudoobscura* (*Paris et al., 2013*). We also gathered data for CTCF in *M. musculus domesticus* (C57BL/6) (*Schmidt et al., 2012*), *R. norvegicus* (*Schmidt et al., 2012*), *H. sapiens* (*Schmidt et al., 2012*), *C. familiaris* (*Schmidt et al., 2012*), *D. melanogaster* (*Ni et al., 2012*), *D. simulans* (*Ni et al., 2012*), *D. yakuba* (*Ni et al., 2012*) and *D. pseudoobscura* (*Ni et al., 2012*). Accession numbers corresponding to the specific experimental replicates and control samples are listed in *Figure 4—source data 1*.

For motif analyses (*Figure 5a,b*; *Figure 5—figure supplement 1*), we gathered known position-weight matrixes from the JASPAR database (*Mathelier et al., 2014*) and the Fly Factor survey (*Zhu et al., 2011*). We focused on the motifs corresponding to Twist and Giant in *D. melanogaster*, to CEBPA, HNF4A and FOXA1 in *M. musculus domesticus*, and on a set of 12 other motifs corresponding to GSTFs conserved across mammals and insects. This set was constructed by downloading all Core A vertebrata motifs from JASPAR (*Mathelier et al., 2014*), identifying those corresponding to conserved GSTFs with one-to-one orthologs between *M. musculus domesticus* and *D. melanogaster* using COMPARA (*Vilella et al., 2009*), and filtering the list down to those 12 instances where a position-weight matrix was also described in Fly Factor (*Zhu et al., 2011*) and were not already analyzed.

## Comparing evolutionary rates

We developed a statistical framework to compare evolutionary rates between lineages, and implemented it in R (*Development Core Team, 2011*). This framework takes as inputs: measures of pairwise cross-species similarity (e.g. correlation of gene expression or sequence conservation), pairwise cross-species evolutionary distances and lineage labels. Conceptually, the framework estimates both a statistical significance and an effect size to determine whether rates of evolutionary divergence are indistinguishable or different between lineages (*Figure 1*).

In practice, we model evolutionary divergence by an exponential decay in log-linear space. First, the nls function in R is applied to the log-transformed cross-species similarity data as a function of evolutionary distances to derive the following linear models:

- a lineage-naïve model that estimates a shared intercept and slope for all the data without specifying the lineage labels
- a lineage-aware model that estimates a shared intercept for all the data and lineage-specific slopes based on lineage labels
- lineage-specific models that estimate intercept and slope individually for each lineage

Second, an R function written in-house to handle nls model structures estimates the significance level of an ANOVA with a likelihood ratio test comparing the lineage-naïve and the lineage-aware model. Third, we define the effect size as the predicted absolute difference in similarity between lineage pairs after 100 Myrs of divergence as estimated from the lineage-specific models. We consider that the framework detected a difference between evolutionary divergence rates when the significance level is <0.05 and the effect size is >5%.

We chose to use an exponential decay function because it is the simplest evolutionary model that fit all our input measures of cross-species similarity reasonably well. We chose to model the exponential decay in log-linear space because we noted that a simple exponential decay in linear space failed to capture the conservation observed between distant species (mouse versus human at 91 Myrs and dog at 97.4 Myrs) when analyzing the evolutionary dynamics of GSTF binding (*Figure 4*) and motif retention (*Figure 5*). We hypothesize that these data layers likely follow a more complex decay model, but we did not want to explore this with our current data set to avoid over-fitting.

The power of our statistical framework was assessed by simulating data for two lineages with measure of cross-species similarity decaying exponentially at different rates over time (*Figure 1—figure supplement 2*). We fixed one lineage to decay at set rates: −0.007, −0.005 and −0.003. We fixed the second lineage to be faster by a range of given differences. Over 1000 simulations, we sampled two values from a normal distribution centered on the expected values from the set exponential decay rates corresponding to the evolutionary distances shown in *Figure 4b*, with standard deviations set at 0.5% or 5%. Our framework detected an absolute rate difference of 0.001 in 39.3% of simulations and an absolute rate difference of 0.003 in 88.9% of simulations when the standard deviation was high (5%). When the standard deviation was low (0.5%), our framework detected an absolute rate difference of 0.001 in 25.7% of simulations and an absolute rate difference of 0.003 in 100% of simulations.

## Gene expression evolutionary rates (related to *Figure 3*)

Analysis of gene expression evolutionary rates was performed in four steps. First, we preprocessed the raw RNA sequencing data downloaded for public data sources. Second, we quantified the abundance of all annotated transcripts corresponding to protein-coding genes. Third, we estimated cross-species similarity by correlating transcript abundances at the genome-scale. Finally, we used these cross-species similarity estimates as input to our statistical framework to evaluate a common model against a lineage-aware model.

RNA sequencing data were first preprocessed using FastQC (www.bioinformatics.babraham.ac.uk/projects/fastqc/) and Trimmomatic (*Bolger et al., 2014*). In order to quantify transcript abundances, we then used the program Sailfish (*Patro et al., 2014*) (1) to build transcriptome indices for each species using the transcriptome sequences described above, using the parameters '-p 8 -k 20'; and (2) to quantify transcript abundance using the transcriptome indices with the parameters '-p 8 -l "T=PE:O=><:S=U"' for samples with paired-end reads and '-p 8 –l "T=SE:S=U"' for samples with single-end reads. The bias-corrected transcripts per million (TPM) abundances estimated by Sailfish were then summed over the transcripts corresponding to the same gene locus.

To estimate cross-species similarities in gene expression levels, for each lineage, we used R (*Development Core Team, 2011*) to build a matrix containing the gene expression values for all the protein-coding genes of the reference organism and their one-to-one orthologs across other organisms within each lineage. We discarded instances where the abundance of a particular gene locus was less than or equal to 5 TPM. We then calculated the Spearman's rank correlation for the expression of all genes between the reference and all other organisms within each lineage and plotted these correlations as against the evolutionary distance separating each organism pair (*Figure 3*). We also repeated the calculations using Kendall's rank correlation coefficient and Pearson's product-moment correlation on $log_2$-transformed expression values (*Figure 3—figure supplement 1a,b*). Finally, we calculated Spearman's correlations among all genes including those with less than 5 TPM (*Figure 3—figure supplement 1c*). All these scenarios were evaluated using our statistical framework. None indicated that a lineage-aware model described the data better than a common model.

## GSTF occupancy – segment-resolution (related to *Figure 4a–d*)

The first step of all our occupancy analyses was to align the ChIP-seq reads to the corresponding genomes in order to obtain occupancy profiles (*Figure 4a*). For each accession (*Figure 4—source data 1*), the sequencing reads were aligned to reference genomes using Bowtie2 version 2.2.4 (*Langmead and Salzberg, 2012*) with the parameters '-very-sensitive -N 1.' Reads containing the 'XS:' field (multi-mappers) were removed. Reads having the same start site were presumed to be PCR duplicates and removed using the 'rmdup' command of SAMtools version 1.1 (*Li et al., 2009*). The filtered reads were then converted to tagAlign format. The tagAlign files corresponding to CEBPA, HNF4A, FOXA1, Twist and Giant were then processed using 108 different segment-resolution methods and one nucleotide-resolution method; the tagAlign files corresponding to CTCF were only processed using the nucleotide-resolution method. The nucleotide-resolution method is described below and relates to *Figure 4e,f*.

The aim of our segment-resolution analyses was to examine how robust the evolution of GSTF binding patterns was across 108 different analysis pipelines (*Figure 4a–d*). We implemented all these pipelines, which follow the same general framework and differ only in the choice of 5 parameters, described and underlined below.

First, the occupancy profiles in the tagAlign files were discretized into candidate occupied segments using a peak caller algorithm that aims at identifying segments where the ChIP sample is enriched in reads relative to the control sample. We implemented two peak callers: MACS version 2 (M) (*Zhang et al., 2008*) and SPP (S) (*Kharchenko et al., 2008*).

The occupied segments were then selected from the candidate set using a quality filter: stringent (S), lenient (L) or asymmetric (A). When using MACS2 (*Zhang et al., 2008*) as a peak caller, lenient segments were called using a p-value cutoff of $10^{-5}$ (default) and merged across replicates when available using the merge function in BEDTools (*Quinlan and Hall, 2010*). Stringent segments were called using a p-value cutoff of $10^{-22}$ and intersected across replicates when replicates were available. The intersection procedure, inspired from (*Stefflova et al., 2013*), used BEDTools (*Quinlan and Hall, 2010*) to implement the following two steps: (1) merge the two replicates and (2) select the merged segments corresponding to at least one segment in each original replicate. When using SPP (*Kharchenko et al., 2008*) as a peak caller, lenient segments were called using a q-value of $10^{-2}$ (default), and merged across replicates when available (*Quinlan and Hall, 2010*). Stringent segments were called by selecting all candidate segments assigned to the lowest possible q-value in the sample, then intersected across replicates when available using the same intersection procedure. The asymmetric quality filter, inspired by (*Bardet et al., 2012*; *He et al., 2011b*), indicates that segments were called stringently in the reference species and leniently in the other organism.

The coordinates of the occupied segments called in the reference organism were projected onto the other organism's genome using the LiftOver tool from the UCSC genome browser (*Rosenbloom et al., 2015*) and specifying a sequence similarity filter through the minMatch parameter. We used 3 different minMatch thresholds: stringent (S: 0.95 default), lenient (L: 0.5), and none (N: 0.001).

After cross-species coordinate projection, a reference subset was chosen to define the set of reference-occupied segments that would be further analyzed. Three choices were implemented: all reference-occupied segments independently of whether they map to any other species (A); for each pair of species, only reference-occupied segments with a homolog in the second species (P); only reference-occupied segments that had homologs across all the other species considered within the lineage (S).

The projected coordinates of the reference subset were then overlapped with the coordinates of the occupied segments in the other species using the intersect function in BEDTools (*Quinlan and Hall, 2010*). The overlap requirement was either lenient (L; default parameter of 1 bp) or stringent (S; required a reciprocal overlap of half of the segments length: '-f 0.5 -r').

We systematically executed all combinations of the aforementioned 2 peak callers, 3 quality filters, 3 sequence similarity filters, 3 reference subsets, and 2 overlap requirements, yielding a total of 108 pipelines. The output of each pipeline was the fraction of reference subset segments that overlapped segments occupied in the others species (i.e. segments retaining occupancy between the two species). This output was used as a cross-species similarity measure for GSTF binding patterns. We analyzed these similarity measures for 6 pairs of GSTFs (Twist and Giant were each compared to

FOXA1, CEBPA and HNF4A) using our statistical framework. Two GSTFs were considered to diverge differently from each other over time when 1) the significance of the test was less than 0.05 and 2) the effect size was greater than 5%. In summary we found that the choice of parameters greatly influenced what the evolutionary dynamics of a given GSTF looked like (*Figure 4b,c*) but that in general the rate of divergence of mammal and insects GSTFs were statistically indistinguishable (*Figure 4d*). The results of these tests for all GSTF pairs considered across 108 pipelines are reported in *Figure 4—source data 2* and summarized as pie-charts in *Figure 4*. Observations about general trends of parameters and evolutionary divergence are further elaborated in *Figure 4—figure supplement 1*.

As a control we also conducted an analysis between FOXA1 and CEBPA since FOXA1 lacks data past 20 Myrs of evolutionary divergence, whereas for all others GSTFs we have broader coverage across the 100 Myrs range. We applied the same statistical framework to the within-lineage comparison between FOXA1 and CEBPA and detected that FOXA1 evolves faster than CEBPA in 74/108 instances. We believe that most of these detected differences are artifacts because the conservation of binding patterns for FOXA1 and CEBPA is in fact highly correlated throughout all combinations of parameters when restricting analyses to data points up to 20 Myrs (Pearson's $r$ = 0.96). We suspect that this type of artifact also affects the results of comparing FOXA1 with Twist or Giant (*Figure 4d*).

## GSTF occupancy – nucleotide-resolution (related to *Figure 4e,f*)

In order to compare occupancy profiles directly without discretizing them into occupied segments and unoccupied segments, we correlated sets of imputed fragment density vectors across species. The inputs to this method were the tagAlign files described above. To generate these vectors we first estimated the mean fragment size using a method adapted from (*Kharchenko et al., 2008*), whereby the mean fragment size is computed as the number of base pairs of offset between the positive and negative strands that maximizes the Pearson's correlation coefficient of their mapped read density. We used a modified approach that considered only the density of 5' read start sites on each strand, rather than the density of the entire read. The first peak of the cross-correlation values was identified by approximating the first derivative by the finite difference method, smoothing the derivative values with a Gaussian kernel of bandwidth 10, and identifying the first downward zero-crossing of the curve. This position was used as the estimated mean fragment size $L$. We created imputed fragments by extending each read start site by $L$ base pairs in the 3' direction. We then calculated a fragment density vector for each chromosome as the number of such imputed fragments that overlap each genomic position. When multiple replicates were available, replicates were merged by adding the fragment density vectors.

In order to minimize bias introduced by the presence of unmappable regions, we implemented a masking scheme that adaptively normalizes each dataset depending on the read length and estimated fragment size of each sequencing run. First, all possible error-free reads of a given length were generated synthetically and aligned back to the genome using Bowtie2 2.2.4 with the following parameters: '-r -N 0 -D 0 -R 0 –dpad 0 –score-min "C,0,-1"'. Any multi-mapping reads with the 'XS:' flag were removed and the 5' and 3'-most positions of the remaining read alignments recorded. The imputed fragment densities computed from the ChIP data were then normalized by dividing the density at each position by the fraction of positions within $L$ base pairs upstream that were covered by the start site (5' for positive-strand density and 3' for negative-strand density) of a uniquely-mapped genomic read. Positions with 0 uniquely-mappable read start sites within $L$ base pairs upstream were excluded from further analysis.

In order to compare between species, we transferred data from query organisms to the reference genome using the one-to-one filtered chain files described previously, and calculated the Pearson's correlation between the concatenated chromosome vectors of reference and reference-mapped query data. The evolution of the correlation was modeled and compared using the statistical framework described above.

## Genome sequence evolutionary rates (related to *Figure 2* and *Figure 5c,d*)

We calculated the percentage of randomly sampled segments retaining homology. Within the genomes of the reference species, we delineated the boundaries of the regions from which to sample: whole genome (*Figure 2*; *Figure 2—figure supplement 1*), intergenic regions in accessible chromatin and intergenic regions in inaccessible chromatin (*Figure 5*; *Figure 5—figure supplement 2*). We used the BEDTools shuffle command (*Quinlan and Hall, 2010*) to randomize the locations of 5000 segments of 75 bp length within the delineated boundaries using the option '-noOverlapping'. The resulting 5000 shuffled segments were then mapped across species using the LiftOver tool with minMatch parameter 0.001 (*Rosenbloom et al., 2015*). We then calculated the percentage of segments that were successfully mapped (i.e. retained homology), excluding segments that mapped to a region longer than 1000 bp. The entire simulation was repeated 20 times, starting each time with different sets of 5000 segments. The percentages of segments retaining homology were recorded for each of the 20 simulations, and averaged for each pair of species. These averages were plotted and used as inputs for our statistical framework. Varying the minMatch parameter of the LiftOver tool to 0.5 and segment length to 150 bp allowed us to verify that the observed trends were robust to sequence similarity thresholds and length sampled (*Figure 2—figure supplement 2*; *Figure 5—figure supplement 2*).

## Nucleotide substitution rate within retained genomic segments (related to *Figure 2—figure supplement 1*)

The nucleotide sequences of the genomic segments from *Figure 2* that retained enough homology to undergo a pairwise alignment were extracted using the getfasta function of BEDTools (*Quinlan and Hall, 2010*). These sequences were then pairwise aligned using EMBOSS suite's implementation of Smith-Waterman local alignment (*Rice et al., 2000*). Default values for gap open penalty (10), gap extend penalty (0.5) and scoring matrix (EDNAFULL) were used to dynamically choose the best local alignment between reference and query sequences. For each cross-species comparison, we calculated the average percent identity of the ungapped alignments of all the segments across 20 randomizations. This procedure yielded values similar to those described previously for the mouse / human (*Waterston et al., 2002*) and *D. melanogaster / D. pseudoobscura* comparisons (*Richards et al., 2005*). The average percent identity of ungapped alignments were used as inputs for our statistical framework, revealing that a model that incorporates lineage labels significantly improved fit to the data relative to a common model (*P<0.05*; *Figure 2—figure supplement 1*).

## Motif evolutionary rates (related to *Figure 5a,b*)

Using the FIMO tool (*Grant et al., 2011*) in the MEME suite (*Bailey et al., 2009*), the genomes of *D. melanogaster* and *M. musculus domesticus* were scanned for matches to experimentally-determined position-weight matrixes corresponding to the GSTFs of interest. Motif matches were called significant according to the default threshold of FIMO, $P<10^{-4}$. The genomic coordinates of significant motif matches were mapped to the other species within the same lineage using LiftOver (minMatch 0.001). The corresponding coordinates (*Mapped*) were then extended by 50 bp, and the resulting segments were scanned for motif occurrence (*Mappedwithmotif*). In order to estimate background expectation, we randomly shuffled the locations of the *Mapped* segments and scanned these shuffled segments for motifs (*ShuffledMappedwithmotif*). The percentage of motifs retained relative to background was calculated as:

$$F = \frac{Mappedwithmotif - ShuffledMappedwithmotif}{Mapped} * 100$$

The percentages *F* were then used as measures of cross-species similarity to estimate whether a lineage-aware model would describe the evolution of DNA binding motifs better than a common model (*Figure 5—figure supplement 1*).

## Acknowledgements

This work was supported by the National Institutes of Health: R01 GM084279 and P50 GM085764 awarded to TI, T32 GM008666 awarded to TW and the Pathway to Independence K99 GM108865 awarded to A-RC. The authors are grateful to Drs. Pollard KS, Wittkopp PJ, Burke M, Charloteaux B and Coolon JD for review of the manuscript prior to submission and to the editors and reviewers for their help in improving the manuscript after submission.

## Additional information

### Funding

| Funder | Grant reference number | Author |
|---|---|---|
| National Institute of General Medical Sciences | R01 GM084279 | Trey Ideker |
| National Institute of General Medical Sciences | P50 GM085764 | Trey Ideker |
| National Institute of General Medical Sciences | T32 GM008666 | Tina Wang |
| National Institute of General Medical Sciences | K99 GM108865 | Anne-Ruxandra Carvunis |

The funders had no role in study design, data collection and interpretation, or the decision to submit the work for publication.

### Author contributions

A-RC, TW, DS, AY, JC, JFK, TI, Conception and design, Analysis and interpretation of data, Drafting or revising the article

### Author ORCIDs

Anne-Ruxandra Carvunis, http://orcid.org/0000-0002-6474-6413

## Additional files

### Supplementary files

- Supplementary file 1. Published ChIP-seq studies comparing binding locations of GSTFs in closely related metazoans used different technical methodologies to estimate divergence rates.

- Supplementary file 2. Parameters used to build chain files among vertebrate genomes.

### Major datasets

The following previously published datasets were used:

| Author(s) | Year | Dataset title | Dataset URL | Database, license, and accessibility information |
|---|---|---|---|---|
| Hao P, Melia T, Sugathan A, Waxman DJ | 2013 | RNA-seq analysis of gene expression in male and female mouse liver | http://www.ncbi.nlm.nih.gov/geo/query/acc.cgi?acc=GSE48109 | Publicly available at the NCBI Gene Expression Omnibus (Accession no: GSE48109). |

| | | | | |
|---|---|---|---|---|
| Brawand D, Soumillon M, Necsulea A, Julien P, Csárdi G, Harrigan P, Weier M, Liechti A, Aximu-Petri A, Kircher M, Albert FW, Zeller U, Khaitovich P, Grützner F, Bergmann S, Nielsen R, Pääbo S, Kaessmann H | 2011 | The evolution of gene expression levels in mammalian organs | http://sra.dnanexus.com/studies/SRP007412 | Publicly available at the Sequence Read Archive + (Accession no: SRP007412). |
| Malone JH, Artieri CG, Sturgill D, Zhang Y, Oliver B | 2011 | mRNA-Seq of whole flies from Drosophila | http://www.ncbi.nlm.nih.gov/geo/query/acc.cgi?acc=GSE28078 | Publicly available at the NCBI Gene Expression Omnibus (Accession no: GSE28078). |
| Negre N, Bild NA, White KP | 2009 | Genome-wide transcriptome sequencing at different stages of Drosophila development, RNA-seq | http://www.ncbi.nlm.nih.gov/geo/query/acc.cgi?acc=GSE18068 | Publicly available at the NCBI Gene Expression Omnibus (Accession no: GSE18068). |
| Goncalves A, Leigh-Brown S, Thybert D, Stefflova K, Turro E, Flicek P, Brazma A, Odom DT, Marioni JC | 2012 | Compensatory cis-trans regulation of mouse liver gene expression | http://sra.dnanexus.com/studies/ERP001401 | Publicly available at the Sequence Read Archive + (Accession no: ERP001401). |
| Wong ES, Thybert D, Schmitt BM, Stefflova K, Odom DT, Flicek P | 2015 | RNA-seq data from liver tissues of Mus spretus (SPRET/EiJ ) and Mus caroli (Caroli/EiJ) | http://www.ebi.ac.uk/ena/data/view/ERP005559 | Publicly available at the EBI European Nucleotide Archive (Accession no: ERP005559 |
| Wang C, Auerbach SS, Shi L, Tong W | 2014 | SEQC Toxicogenomics Study: RNA-Seq data set | http://www.ncbi.nlm.nih.gov/geo/query/acc.cgi?acc=GSE55347 | Publicly available at the NCBI Gene Expression Omnibus (Accession no: GSE55347). |
| Lin S, Lin Y, Nery JR, Urich MA, Breschi A, Davis CA, Dobin A, Zaleski C, Beer MA, Chapman WC, Gingeras TR, Ecker JR, Snyder MP | 2014 | RNA-seq of human liver tissue | https://www.encodeproject.org/experiments/ENCSR085HNI/ | ENCSR085HNI |
| Malone JH, Artieri CG, Sturgill D, Zhang Y, Oliver B | 2011 | mRNA-Seq of whole flies from Drosophila | http://www.ncbi.nlm.nih.gov/geo/query/acc.cgi?acc=GSE28078 | Publicly available at the NCBI Gene Expression Omnibus (Accession no: GSE28078). |
| Monson MS, Settlage RE, McMahon KW, Mendoza KM, Rawal S, El-Nezami HS, Coulombe RA, Reed KM | 2014 | Transcriptome responses to dietary aflatoxin B1 exposure in domestic turkey | http://trace.ddbj.nig.ac.jp/DRASearch/study?acc=SRP042724 | SRP042724 |
| Li N, Huang Y | 2013 | Genome and transcriptome of the duck | http://www.ncbi.nlm.nih.gov/geo/query/acc.cgi?acc=GSE22967 | Publicly available at the NCBI Gene Expression Omnibus (Accession no: GSE22967). |
| Ellegren H, Smeds L, Burri R, Olason PI, Backström N, Kawakami T, Künstner A, Mäkinen H, Nadachowska-Brzyska K, Qvarnström A, Uebbing S, Wolf JB | 2012 | Whole genome sequencing and de novo assembly of the collared flycatcher | http://sra.dnanexus.com/?result_type=Study&show=25&q=ERP001377 | Publicly available at the Sequence Read Archive + (Accession no: ERP001377). |

| | | | | |
|---|---|---|---|---|
| Stefflova K, Thybert D, Wilson MD, Streeter I, Aleksic J, Karagianni P, Brazma A, Adams DJ, Talianidis I, Marioni JC, Flicek P, Odom DT | 2013 | Transcription Factor binding evolution in five mouse species | http://sra.dnanexus.com/?result_type=Study&show=25&q=+ERP002082 | Publicly available at the Sequence Read Archive + (Accession no: ERP002082). |
| Schmidt D, Wilson MD, Ballester B, Schwalie PC, Brown GD, Marshall A, Kutter C, Watt S, Martinez-Jimenez CP, Mackay S, Talianidis I, Flicek P, Odom DT | 2010 | CEBPA binding in five vertebrates | http://sra.dnanexus.com/?result_type=Study&show=25&q=ERP000054 | Publicly available at the Sequence Read Archive + (Accession no: ERP000054). |
| Bradley RK, Li X, Eisen MB | 2010 | Binding site turnover produces pervasive quantitative changes in TF binding between closely related Drosophila species | http://www.ncbi.nlm.nih.gov/geo/query/acc.cgi?acc=GSE20369 | Publicly available at the NCBI Gene Expression Omnibus (Accession no: GSE20369). |
| Paris M, Kaplan T, Li XY, Villalta JE, Lott SE, Eisen MB | 2013 | Extensive divergence of transcription factor binding in Drosophila embryos with highly conserved gene expression | http://www.ncbi.nlm.nih.gov/geo/query/acc.cgi?acc=GSE50773 | Publicly available at the NCBI Gene Expression Omnibus (Accession no: GSE50773). |
| He Q, Bardet AF, Patton B, Purvis J, Johnston J, Paulson A, Gogol M, Stark A, Zeitlinger J | 2011 | cross_species | http://sra.dnanexus.com/studies/ERP000357 | Publicly available at the Sequence Read Archive + (Accession no: ERP000357). |
| Schmidt D, Schwalie PC, Wilson MD, Ballester B, Gonçalves A, Kutter C, Brown GD, Marshall A, Flicek P, Odom DT | 2012 | Waves of Retrotransposon Expansion Remodel Genome Organization and CTCF Binding in Multiple Mammalian Lineages | http://sra.dnanexus.com/?result_type=Study&show=25&q=ERP000395 | Publicly available at the Sequence Read Archive + (Accession no: ERP000395). |
| Ni X, Zhang YE, Nègre N, Chen S, Long M, White KP | 2012 | Genome-wide comparative ChIP-seq data of CTCF and RNA-seq data in Drosophila white prepupa on Illumina Genome Analyzer | http://sra.dnanexus.com/?result_type=Study&show=25&q=SRP003653 | Publicly available at the Sequence Read Archive + (Accession no: SRP003653 |
| Stefflova K, Thybert D, Wilson MD, Streeter I, Aleksic J, Karagianni P, Brazma A, Adams DJ, Talianidis I, Marioni JC, Flicek P, Odom DT | 2013 | Transcription Factor binding evolution in five mouse species | http://sra.dnanexus.com/?result_type=Study&show=25&q=+ERP002078 | Publicly available at the Sequence Read Archive + (Accession no: ERP002078). |

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
