## [Decision Letter]

Thank you for submitting your work entitled "Evidence for a common evolutionary rate in metazoan transcriptional networks" for consideration by *eLife*. Your article has been reviewed by three peer reviewers, one of whom is a member of our Board of Reviewing Editors (Duncan Odom). The evaluation has been overseen by Naama Barkai as the Senior Editor.

The reviewers have discussed the reviews with one another and the Reviewing editor has drafted this decision to help you prepare a revised submission.

Summary:

This was unanimously evaluated as an important analytical contribution to regulatory evolution, and will help rectify a number of discordant results currently in the literature. Carvunis et al. confirm that the rate of gene expression evolution is comparatively stable in both fruit flies and mammalian clades, and newly show that using a uniform analysis approach strongly suggests that underlying TF binding evolution also appears to occur at a comparable rate. They provide some evidence towards the hypothesis that this rate is mediated by the different rates of change found in euchromatic versus heterochromatic components of the mammalian genome.

Essential revisions:

1) The methods should be more carefully documented and the main text and supporting materials sections better cross-referenced (Reviewer #1).

2) The Discussion should be somewhat increased, and the major points brought up by all three reviewers addressed.

3) A few of the results should be revisited with new analysis (see Reviewer #3).

Reviewer #1:

This is an extremely important manuscript in regulatory evolution that, in essence, harmonizes apparently contradictory results from a number of previously published studies by using a carefully designed and uniform analysis methodology. There has been considerable debate around how rapidly transcription and transcriptional regulation evolve between species. In particular, fruit flies and mammals have been reported to have surprisingly different tempos of gene-specific TF binding divergence. Carvunis et al. extract the raw data from a number of studies, and then apply rigorously a combination of analysis protocols with many combinations of widely used tools. They reveal that when analyzed together, gene expression and TF binding evolve similarly in drosophila and mammals. Finally and importantly, Carvunis et al. reveal that a likely culprit for the disconnect between rapid sequence evolution of mammals and the slower SPTF binding and gene expression evolution is the rapid divergence of chromatin inaccessible DNA in mammals. This last point is the key intellectual (as opposed to technical/analytical) insight within this paper.

Aside from more clearly outlined computational handling and a few minor revisions I suggest for consideration below, this straightforward and timely paper will be an excellent contribution to the ongoing discussion of evolutionary genetics.

Major considerations:

1) Materials and methods. Starting from paragraph two of the Results, the authors should direct the reader to the Methods section, and write a concise, but carefully laid out, description of how each step was performed. For instance from paragraph two of the Results, the inter-species normalization of gene expression is a notoriously tricky bit of analysis, per the Brawand et al. Nature 2011 study. I think this really must be carefully walked through. Similarly for all the other sections later.

2) Points not mentioned in this version that could be additionally dissected in the Discussion include:

The impact of effective breeding sizes on how chromatin accessible and inaccessible DNA is handled.

How the breeding rate and absolute time (MY) could interplay in driving evolution (in other words, 40 MY for flies is a LOT more generations than for mice).

Reviewer #2:

This study draws on published datasets to study the evolution of transcriptional regulation in three clades (mammals, birds, and diptera). The basic finding is that rates of evolutionary divergence in transcript abundance, transcription factor binding site occupancy, open chromatin sequence evolution, and transcription factor motif sequences are similar in the three clades. This result is unexpected, as previous studies have reported rather different rates of evolution for some of these features. The authors use this observation to argue that only a small fraction of the genome is involved in transcriptional regulation.

A notable strength of this study is that the authors applied uniform methods of analysis to similar kinds of data, demonstrating that the differences in evolutionary rates reported in the earlier publications are at least in part an artifact of the different methods of analysis that they used. (The impact of different technology platforms, which is also likely a contributing factor, was not addressed.) These results provide an important cautionary lesson, namely that it is essential to work with closely comparable data and to apply uniform methods of analysis before drawing conclusions about biological differences or similarities based on functional genomic datasets. The study is valuable for this reason alone.

Where it is less successful is in providing new insights:

1) The plots showing similar overall rates of evolution (Figure 3, Figure 4, Figure 5) are striking and I'm convinced that the rates are similar among the three clades. But I have no idea what this tells us about the evolution of transcription. I couldn't find any mention in the Results or Discussion sections about how to interpret the observation of rate constancy.

2) The only conclusion presented in the Discussion concerns the fraction of the genome that regulates transcription. Unfortunately, the bulk of the evolutionary comparisons have no bearing on this conclusion. The only one that does is the rate of sequence evolution within chromatin-accessible regions of the genome after removing core promoters. This comparison is relevant, but not because of the rate constancy among clades. It is relevant because the rate in this fraction of the genome is notably slower than the genome as a whole, and specifically within mammals, which have the greatest proportion of noncoding sequences among the three clades. Thus, most of the rate comparisons among the three clades contribute little or nothing to the conclusion that a small fraction of the genome regulates gene expression.

3) The observation that regulatory sequences evolve more slowly than other noncoding sequences is by no means a novel observation. This has been noted previously not just in Sundaram et al. (2014) (as the authors point out), but in several others, including Yue et al. (2014) and papers from Odom's group and Crawford's group. So the basic conclusion of the study was already apparent. Indeed, this is the logic that Graur et al. (2013) used to argue that the ENCODE team overestimated the fraction of noncoding sequences that are functional.

4) The argument that sequence conservation implies that a small fraction of sites regulate transcription is not as straightforward as implied in the Discussion. The ENCODE team wrote a thoughtful rebuttal of the Graur et al. paper (Kellis et al. 2014 PNAS 111:6131) that should be cited in this regard. Beyond the narrow debate about the ENCODE claims, it has been clear for quite some time that some enhancers and transcription factor binding sites turn over quite rapidly in evolution even though they are known to play a role in transcriptional regulation. See Yue et al. (2014) for a recent example, but there are many others. It is clear that negative selection is not the only evolutionary mechanism that operates on regulatory sequences. Using conservation as a criterion for function will thus overlook some functional sites.

*Reviewer #3:*

Carvunis et al. present an interesting analysis comparing evolutionary rate of genomic sequences and transcriptional regulation, by examining gene expression, transcription factor binding, and open chromatin marks across multiple species of mammals, birds, and drosophila. The authors' analyses recapitulate previous results that sequence gain and loss are much more prominent in mammals than in birds and insects, while rates of gene expression level changes are similar. The authors show that the rates of regulatory changes, such as gain and loss of orthologous TF binding and open chromatin events, are indistinguishable between lineages, providing an answer to conflicting reports on the different rates of evolution for genomic sequences and transcription regulation.

This is a well-written study with important insights. The authors draw a significant conclusion. However, the study has several limitations in their results, reducing their supports to a rather strong conclusion. The authors should either be very clear with the assumptions they made, or presenting stronger and more comprehensive evidence. But I am very excited about this work.

First, the authors use fraction of ortholog sequence retained as a metric to measure rate of genomic sequence evolution. However, the study should consider sequence evolution in the context of rate of substitution, in addition to what the authors already provide. Retained sequences can evolve at different rate, which is directly related to the arguments the authors make.

Second, the authors choose to use the exponential decay model to fit this data. While this is a useful model for evolutionary analysis, this is not the only one. Would their conclusion be solid if they fit the data with a different model?

Third, the species chosen for the study does not provide enough coverage across the 100myrs span. For mammals, there was one data point at 20myrs, and many more data points at either >5myrs, or more than 90myrs. This is a serious concern, because the skewed distribution could potentially bias any model fitting. This is reflected by the TF data for the 20myr comparison. The datapoints at 20myrs were consistently below the fitted curve, often appears quite significantly deviated from the curve.

Lastly, the number of TFs included in the study is too small to support a quite general and significant conclusion. I don't expect many TF data will become available for the study, so perhaps tuning down some of the claims would be a good response.

---

## [Author Response]

*Essential revisions: 1) The methods should be more carefully documented and the main text and supporting materials sections better cross-referenced (Reviewer #1).*

We agree with Reviewer #1 that our methods should be more clearly laid out than they were in the original submission. We thus have more carefully documented our methods and improved cross-referencing throughout the manuscript. For example, the Materials and methods section related to our RNA-seq analyses is now introduced by a succinct paragraph summarizing the various steps of the analysis. Furthermore, each Materials and methods section is now explicitly linked to a figure panel or a set of figure panels.

*2) The Discussion should be somewhat increased, and the major points brought up by all three reviewers addressed.*

The revised manuscript ends with a considerably longer Discussion. Notably, we now examine our results in light of the differences in breeding sizes and breeding rates (generation times) among the lineages examined. We also list a series of limitations that will need to be addressed in future work to determine how universal our results are, and present a more detailed explanation of why our results suggest that only a small fraction of the genome contributes to transcription.

*3) A few of the results should be revisited with new analysis (see Reviewer #3).*

Following Reviewer #3’s suggestions, we have made the following improvements to our analyses:

We have investigated evolution of genome sequences using rate of substitution as a metric in addition to the analyses already provided in the original submission.

To increase the number of data points and the quality of fits in our analyses of ChIP-seq data, we have incorporated an additional species (dog) for which data were available in the literature.

We have improved the sensitivity of our statistical framework.

Detailed explanations of each of these improvements are given below in a point-by-point response to Reviewer #3.

Reviewer #1:

*1) Methods. Starting from paragraph two of the Results, the authors should direct the reader to the Methods section, and write a concise, but carefully laid out, description of how each step was performed. For instance from paragraph two of the Results, the inter-species normalization of gene expression is a notoriously tricky bit of analysis, per the Brawand et al. Nature 2011 study. I think this really must be carefully walked through. Similarly for all the other sections later.*

Following the reviewer’s suggestion, we have added cross-references to the Materials and methods section and to the corresponding supplementary figures and files throughout the manuscript. The description of each step performed for the gene expression analysis is presented in the subsection “Gene expression evolutionary rates (related to Figure 3)” of the revised manuscript and relies on the orthology relationships described in the subsection “Genome and Annotation Sources”. The revised manuscript now includes a paragraph clarifying the steps involved. Similar clarifications were also introduced in other Materials and methods sections. Regarding normalization of RNA-seq data, the algorithm we used for transcript quantification, Sailfish, performs several bias correction steps that are described in the corresponding publication (Patro Nature Biotech, 2014). We note that Brawand and colleagues (Nature, 2011) implemented additional normalization steps in order to search for differentially expressed genes, which we did not attempt in this work.

*2) Points not mentioned in this version that could be additionally dissected in the Discussion include:*

The impact of effective breeding sizes on how chromatin accessible and inaccessible DNA is handled.

*How the breeding rate and absolute time (MY) could interplay in driving evolution (in other words, 40 MY for flies is a LOT more generations than for mice).*

To address this comment, we have significantly lengthened our Discussion. Among other improvements, the revised manuscript discusses our results in light of the differences in breeding sizes and breeding rates (generation times) among the lineages examined.

Reviewer #2:

*1) The plots showing similar overall rates of evolution (Figure 3, Figure 4, Figure 5) are striking and I'm convinced that the rates are similar among the three clades. But I have no idea what this tells us about the evolution of transcription. I couldn't find any mention in the Results or Discussion sections about how to interpret the observation of rate constancy.*

We fully agree with the reviewer that our original submission lacked a thorough Discussion section. Encouraged by this and the other reviewers’ suggestions, we have now included in the revised manuscript a longer Discussion section, including speculative interpretations.

*2) The only conclusion presented in the Discussion concerns the fraction of the genome that regulates transcription. Unfortunately, the bulk of the evolutionary comparisons have no bearing on this conclusion. The only one that does is the rate of sequence evolution within chromatin-accessible regions of the genome after removing core promoters. This comparison is relevant, but not because of the rate constancy among clades. It is relevant because the rate in this fraction of the genome is notably slower than the genome as a whole, and specifically within mammals, which have the greatest proportion of noncoding sequences among the three clades. Thus, most of the rate comparisons among the three clades contribute little or nothing to the conclusion that a small fraction of the genome regulates gene expression.*

To address this comment, we have revised the Discussion section such that the hypothesis that a small fraction of the mammalian genome regulates transcription is presented as one of several possible interpretations.

In the revised Discussion, we claim that the evolutionary rate of transcription is decoupled from the lineage-specific parameters that govern the evolution of genome sequence. We present the hypothesis that a small fraction of the mammalian genome contributes to transcriptional regulation as a plausible explanation given our observations within chromatin-accessible regions. We also present alternative hypotheses where rapidly evolving portions of the bird and insect genomes may make significant contributions to transcriptional regulation.

*3) The observation that regulatory sequences evolve more slowly than other noncoding sequences is by no means a novel observation. This has been noted previously not just in Sundaram et al. (2014) (as the authors point out), but in several others, including Yue et al. (2014) and papers from Odom's group and Crawford's group. So the basic conclusion of the study was already apparent. Indeed, this is the logic that Graur et al. (2013) used to argue that the ENCODE team overestimated the fraction of noncoding sequences that are functional.*

We fully agree with the reviewer that the fact that many regulatory sequences are conserved across species was known before our submission. We specifically quoted Yue et al. (2014) in this context as it pertains specifically to comparing chromatin-accessible sequences to random non-coding regions, similar to what we do in the corresponding paragraph. While the slow divergence of chromatin-accessible regions relative to other non-coding regions in mammals may have been easily predicted from pre-existing literature, there was no previously published indication that such difference would be practically nonexistent in birds and insects, leading to a shared divergence rate across the three clades. We have now added text raising and explaining these points in the manuscript Discussion.

*4) The argument that sequence conservation implies that a small fraction of sites regulate transcription is not as straightforward as implied in the Discussion. The ENCODE team wrote a thoughtful rebuttal of the Graur et al. paper (Kellis et al. 2014 PNAS 111:6131) that should be cited in this regard. Beyond the narrow debate about the ENCODE claims, it has been clear for quite some time that some enhancers and transcription factor binding sites turn over quite rapidly in evolution even though they are known to play a role in transcriptional regulation. See Yue et al. (2014) for a recent example, but there are many others. It is clear that negative selection is not the only evolutionary mechanism that operates on regulatory sequences. Using conservation as a criterion for function will thus overlook some functional sites.*

We wholeheartedly agree with the reviewer that conservation is by no means the only indicator of regulatory function. We have included the Kellis (2014) reference in the Introduction of our revised manuscript. We never meant to imply that only conserved sites can be functional. Our study is showing that, at different levels of the transcription process, evolutionary rates are shared across mammals, birds and insects. Our point is that these rates could not have been predicted from genome-wide patterns of DNA evolution, as they can be slow (transcript levels, chromatin-accessible sequences) or fast (TF binding, motifs), irrespective of how slow or fast the genome evolves as a whole. We have clarified our language in the revised manuscript.

Reviewer #3:

*First, the authors use fraction of ortholog sequence retained as a metric to measure rate of genomic sequence evolution. However, the study should consider sequence evolution in the context of rate of substitution, in addition to what the authors already provide. Retained sequences can evolve at different rate, which is directly related to the arguments the authors make.*

To address this comment, we have measured how much retained sequences changed in nucleotide identity over time. We found that retained sequences were highly conserved at the nucleotide level in mammals, birds and insects. However, insect sequences diverged slightly but significantly faster than bird and mammal sequences. The results of this analysis are shown in Figure 2—figure supplement 1 and discussed in the revised manuscript.

*Second, the authors choose to use the exponential decay model to fit this data. While this is a useful model for evolutionary analysis, this is not the only one. Would their conclusion be solid if they fit the data with a different model?*

To address the reviewer’s comment, we conducted an additional analysis aimed at comparing an exponential decay with a linear decay and with a power law decay. Models with more than two parameters were not investigated due to the higher risk of over-fitting.

To compare the quality of these various decay models, we encoded them in R in a unified regression framework relying on non-linear least squares (nls) – as opposed to the linear least square regression (lm) method used in the original submission. We quantitatively estimated fit qualities across all the data layers and lineages used throughout our manuscript using the AICc criteria, a small-sample-size corrected version of Akaike information criterion. We found that the data are almost always better represented by an exponential decay than by either a linear or a power law decay (as determined by the lowest AICc). The only data type better modelled by a power law decay than by an exponential decay was the nucleotide identity of retained genomic segments, a data type introduced to the manuscript as per the above suggestion of the same reviewer. We felt that introducing additional models that do not adequately fit the majority of the data would not make our results more robust. Therefore, we did not specifically investigate if linear or power law decays would give different results than an exponential decay when comparing lineages with our statistical framework, and we kept the exponential model as the basis of our work.

While performing this analysis, we discovered that model fitting using nls was much more sensitive to detect changes in decay rates than fitting with lm. In spite of this increase in sensitivity, the vast majority of our results and all of our major conclusions remained unchanged. We thus opted to use nls throughout our revised manuscript.

*Third, the species chosen for the study does not provide enough coverage across the 100myrs span. For mammals, there was one data point at 20myrs, and many more data points at either >5myrs, or more than 90myrs. This is a serious concern, because the skewed distribution could potentially bias any model fitting. This is reflected by the TF data for the 20myr comparison. The datapoints at 20myrs were consistently below the fitted curve, often appears quite significantly deviated from the curve.*

We agree that the distribution of mammalian species with fully sequenced genomes across the 100 Myrs range is less than ideal for model fitting. We were able to include a large number of species in some of our analyses (e.g. Figure 2 includes rabbit and guinea pig, separated from mice by 75 and 86 Myrs, respectively), but the lack of ChIP data for species separated from mouse by more than 20 but less than 90 Myrs is indeed a concern (reflected specifically in Figure 4 and Figure 5).

In these specific figure panels, the data points at 20 Myrs (corresponding to mouse-rat comparison) are often below the fitted curve while the data points at 91 Myrs (mouse-human) are reasonably well captured. Given the data presented in the submitted manuscript, it is unclear whether the mouse-rat or the mouse-human comparison is the outlier. To remedy this gap to the best of our ability given the available ChIP data, we have added dog samples to our analyses of HNF4A, CEBPA and CTCF. Since dog is very distant from mouse (97 Myrs), adding these samples improved fit confidence at large evolutionary distances. In addition to this improvement, we also added to the revised manuscript a discussion of the limitations of our study posed by the non-uniform coverage of datasets across evolutionary time.

*Lastly, the number of TFs included in the study is too small to support a quite general and significant conclusion. I don't expect many TF data will become available for the study, so perhaps tuning down some of the claims would be a good response.*

We agree that the number of TFs with occupancy data investigated is too small to make a generalization on TF occupancy evolution, and we have therefore toned down our language in the corresponding paragraphs accordingly.